# Identification of Variants (rs11571707, rs144848, and rs11571769) in the *BRCA2* Gene Associated with Hereditary Breast Cancer in Indigenous Populations of the Brazilian Amazon

**DOI:** 10.3390/genes12020142

**Published:** 2021-01-22

**Authors:** Elizabeth Ayres Fragoso Dobbin, Jéssyca Amanda Gomes Medeiros, Marta Solange Camarinha Ramos Costa, Juliana Carla Gomes Rodrigues, João Farias Guerreiro, José Eduardo Kroll, Sandro José de Souza, Paulo Pimentel de Assumpção, Ândrea Ribeiro-dos-Santos, Sidney Emanuel Batista dos Santos, Rommel Mario Rodríguez Burbano, Marianne Rodrigues Fernandes, Ney Pereira Carneiro dos Santos

**Affiliations:** 1Núcleo de Pesquisas em Oncologia, Universidade Federal do Pará, Belém 66075-110, Brazil; elizabethdobbin7@gmail.com (E.A.F.D.); jessycamandag@gmail.com (J.A.G.M.); martasolange@ig.com.br (M.S.C.R.C.); julianacgrodrigues@gmail.com (J.C.G.R.); assumpcaopp@gmail.com (P.P.d.A.); akelyufpa@gmail.com (Â.R.-d.-S.); sidneysantos@ufpa.br (S.E.B.d.S.); rommelburbano@gmail.com (R.M.R.B.); fernandesmr@yahoo.com.br (M.R.F.); 2Laboratório de Genética Humana e Médica, Instituto de Ciências Biológicas, Universidade Federal do Pará, Belém 66075-110, Brazil; joao.guerreiro53@gmail.com; 3Brain Institute, Universidade Federal do Rio Grande do Norte, Natal 59078-970, Brazil; jkpenga@gmail.com (J.E.K.); sandro@neuro.ufrn.br (S.J.d.S.); 4Hospital Ophir Loyola, Belém 66063-240, Brazil

**Keywords:** hereditary breast cancer, *BRCA1*, *BRCA2*, indigenous populations, Native Americans, Brazil

## Abstract

Estimates show that 5–10% of breast cancer cases are hereditary, caused by genetic variants in autosomal dominant genes; of these, 16% are due to germline mutations in the *BRCA1* and *BRCA2* genes. The comprehension of the mutation profile of these genes in the Brazilian population, particularly in Amazonian Amerindian groups, is scarce. We investigated fifteen polymorphisms in the *BRCA1* and *BRCA2* genes in Amazonian Amerindians and compared the results with the findings of global populations publicly available in the 1000 Genomes Project database. Our study shows that three variants (rs11571769, rs144848, and rs11571707) of the *BRCA2* gene, commonly associated with hereditary breast cancer, had a significantly higher allele frequency in the Amazonian Amerindian individuals in comparison with the African, American, European, and Asian groups analyzed. These data outline the singular genetic profiles of the indigenous population from the Brazilian Amazon region. The knowledge about *BRCA1* and *BRCA2* variants is critical to establish public policies for hereditary breast cancer screening in Amerindian groups and populations admixed with them, such as the Brazilian population.

## 1. Introduction

Breast cancer (BC) is the most common type of cancer in women, excluding non-melanoma cancer. By 2040, more than three million people around the globe will be affected by this cancer type [1]. In Brazil, the National Cancer Institute (INCA) estimates new cases of breast cancer for each year in the 2020–2022 period, this being the most frequent cancer site—excluding non-melanoma cancer—in Brazilian females, including those from the northern region [2].

The *BRCA1* and *BRCA2* genes act as tumor suppressors by maintaining genomic stability and DNA repair [3]. Mutations in these genes are related to a higher susceptibility to BC, which can increase more than 80% of the chances of developing this cancer type during life [4].

About 5–10% of BC cases are due to hereditary factors, which are caused by variants in autosomal dominant genes; of these, 16% can be attributed to germline mutations in the *BRCA1* and *BRCA2* genes [5]. The frequency of allelic variants of these genes varies widely between different population groups. For example, in Ashkenazi Jews, approximately 1 in 40 individuals has one of three specific founder mutations (187delAG or 5385insC in *BRCA1* or 6174delT in *BRCA2*) [6]. Despite the known population variability of variants, little is known about the profile of mutations in the *BRCA1* and *BRCA2* genes in the Brazilian population, particularly in Amazonian Amerindian groups.

According to the Brazilian Institute of Geography and Statistics (IBGE), indigenous people represent 0.47% of the Brazilian population, summing a total of 896,917 individuals [7]; nevertheless, reports about cancer prevalence in this population are still rare. In other countries, such as Australia and New Zealand, records indicate that cancer is the second leading cause of death among indigenous people [8]. There are no studies that assess the impact of genetic variants in Amazonian Amerindian populations and the risk of developing breast cancer.

This study characterizes the molecular profile of the *BRCA1* and *BRCA2* genes by analyzing the exome of Amerindian populations from the Brazilian Amazon. We aimed to describe variants that may affect the development of breast cancer in Amerindian populations and to compare these data with populations of distinct ancestry background.

## 2. Materials and Methods

### 2.1. Study and Reference Populations

The study population is composed of 64 Amerindians from the Amazon region of northern Brazil who represent 12 different Amazonian ethnic groups: five Asurini of the Xingu, seven Arara, six Araweté, 16 Asurini of the Tocantins, eight Awa-Guajá, two Kayapó/Xikrin, five Zo’é, ten Wajãpi, one Karipuna, one Phurere, one Munduruku and two Juruna. All individuals were grouped as the Indigenous group (IND). The Amerindian individuals share no family relationships and do not present breast cancer or cases of breast cancer in their families. The genetic ancestry was obtained through a panel of 64 informative ancestry markers (IAM), as described by Ramos et al. 2016 [9]. Additional information about this population can be found in [10].

All participants of the study and their ethnic group leaders signed a free-informed consent. The recruitment period for participants was from September 2017 to December 2018. The study was approved by the National Committee for Ethics in Research (CONEP) and the Research Ethics Committee of the UFPA Tropical Medicine Center, under CAAE number 20654313.6.0000.5172.

We compared our results with those of populations from other countries available at the phase 3 release of the 1000 Genomes Database (available at http://www.1000genomes.org).

These populations include 5203 individuals of African (AFR), 5789 of American (AMR), 4327 of East Asian (EAS), 33,370 of European (EUR), and 8256 of South Asian (SAS) descent. As indicated in [11], for the samples with ancestry from Europe, East Asia, and South Asia, populations across the geographic range had about 1% FST; the populations from Africa are related to the Yoruba and are therefore not comprehensive within Africa; for populations in the Americas, the samples are from two populations with primarily African and European ancestry (people with African Ancestry in southwest USA (ASW) and the African Caribbean in Barbados (ACB) and four populations (People with Mexican Ancestry in Los Angeles, CA, USA (MXL), Colombians in Medellin, Colombia (CLM), Puerto Ricans in Puerto Rico (PUR), Peruvians in Lima, Peru (PEL) with a wide range of European, African, and indigenous American ancestry chosen to represent the wide variation in ancestry proportions observed in North, Central, and South America.

### 2.2. Extraction of the DNA and Preparation of the Exome Library

The DNA extraction was performed by the phenol-chloroform method [12]. The quantification and integrity of genetic material were analyzed by a Nanodrop-8000 spectrophotometer (Thermo Fisher Scientific Inc., Wilmington, DE, USA) and electrophoresis in 2% agarose gel, respectively.

The exome libraries were prepared using the Nextera Rapid Capture Exome (Illumina^®^, San Diego, CA, USA) and SureSelect Human All Exon V6 (Agilent) kits. The sequencing reactions were run in the NextSeq 500^®^ platform (Illumina^®^, San Diego, CA, USA) using the NextSeq 500 High-output v2 300 cycle kit (Illumina^®^, San Diego, CA, USA).

### 2.3. Bioinformatic Analysis

The bioinformatic analysis was performed as previously described in Rodrigues and colleagues [10].

### 2.4. Statistical Analyses

The allele frequencies of the IND populations were obtained by gene counting and compared with the other study populations (AFR, EUR, AMR, EAS, and SAS). Fisher’s exact test was used to test the difference in frequencies between the populations. A *p*-value ≤ 0.05 was considered significant. The interpopulation variability of the polymorphisms was assessed using Wright’s fixation index (FST). All analyses were run in RStudio v.3.5.1.

### 2.5. Selection of Variants

The selection of variants for subsequent analyses was based on two main criteria: (a) minimum of 10 reads of coverage (fastx_tools v.0.13-http://hannonlab.cshl.edu/fastx_toolkit/); (b) the variant impact should be either modifier, moderate or high, according to SNPeff classification (https://pcingola.github.io/SnpEff/). A total of 41 variants were found in the *BRCA1* and *BRCA2* genes, which are described in Appendix A. The analyses were directed to 15 variants that met all specifications of the selection criteria.

## 3. Results

Fifteen variants were identified, eight in the *BRCA1* gene and seven in the *BRCA2* gene, in the individuals analyzed (Table 1). This table contains the characteristics of these variants, including their reference number, chromosomal region, nucleotide exchange, impact predicted by the SNPeff software, and the allele frequency referring to the IND population and the five continental populations present in the 1000 Genomes platform (AFR, AMR, EAS, EUR, and SAS). Among the selected polymorphisms, five have predicted impact as modifier and ten as moderate.

The multidimensional scale analysis (MDS) using the FST values (Appendix A) for each pairwise comparison for the 15 variants in the *BRCA1* and *BRCA2* genes revealed the existence of three major groups (Figure 1): the African population (AFR) is completely isolated, showing greater genetic diversity; the European (EUR), East Asian (EAS) and South Asian (SAS) populations have grouped in the top right corner; the third group was composed of the American population (AMR) and the Amerindians of this study (IND).

In addition, the frequencies of these 15 variants were compared with the global populations using Fisher’s exact test (Table 2). The molecular profile of the indigenous individuals of this study varies considerably from African, American, and European populations.

The EUR population stands out as the one that most presented variants with significant differences (12 out of 15) to the Amerindian population; of which six were in the *BRCA1* genes (rs799923, rs16941, rs16942, rs1799966, rs1799949, and rs799917) and six in the *BRCA2* gene (rs11571769, rs144848, rs1799943, rs1799944, rs11571707, and rs766173).

Regarding the AFR population, six polymorphisms were found to be significantly divergent in the *BRCA1* gene (rs16941, rs16942, rs1799966, rs799905, rs1799949, and rs799917) and five in the *BRCA2* gene (rs11571769, rs144848, rs1799944, rs11571707, and rs766173), summing a total of 11 significantly different variants from the IND population.

The American population (AMR) presented 10 statistically different polymorphisms in relation to the IND population: six in the *BRCA1* gene (rs799923, rs16941, rs16942, rs1799966, rs1799949 and rs799917) and four in the *BRCA2* gene (rs11571769, rs144848, rs1799944 and rs11571).

Moreover, the comparison between IND and SAS and EAS populations showed the same number of significantly different SNPs (6 out of 15). Two variants in the *BRCA1* gene (rs12516 and rs799917) and four in the *BRCA2* gene (rs11571769, rs144848, rs1799943, and rs11571707) were divergent in the EAS population. The SAS population also showed two divergent polymorphisms in the *BRCA1* gene (rs799923 and rs12516) and four in the *BRCA2* gene (rs11571769, rs144848, rs1799943, and rs11571707).

It is worth mentioning that three variants in the *BRCA2* gene (rs11571769, rs144848, and rs11571707) presented a unique frequency in the Amazonian Amerindian populations, diverging from that observed in all other evaluated populations. The distribution of the mutant alleles in the indigenous ethnicities can be consulted in Table 3. No mutant homozygotes were observed for rs11571769 in the IND group, whereas for the rs11571707 and the rs144848 variants, 8 and 16 individuals had homozygous mutant genotypes, respectively.

The allele frequency of these polymorphisms is substantially higher in indigenous individuals when compared to individuals of other ancestries. The rs11571707 (NM_000059.4:c.7469T>C; NP_000050.3:p.Ile2490Thr) showed the greatest allele frequency in the study subjects (32.03%-41 mutant alleles) in contrast to the world populations (except the AMR) where the frequencies are less than 1%. Similarly, the rs144848 (NM_000059.4:c.1114A>C, NP_000050.3:p.Asn372His) also had a significantly high-frequency in indigenous people (50.78%-65 mutant alleles), whereas the other populations showed a mean frequency lower than 27%. Additionally, the rs11571769 (NM_000059.4:c.8851G>A NP_000050.3:p.Ala2951Thr) showed a frequency of 11.1% (14 mutant alleles) in the IND individuals in contrast to a mean frequency of 1% in the other populations.

## 4. Discussion

Recently, in Latin America and Brazil, cancer has been growing gradually among the indigenous people [8,13,14]. Around 10% of indigenous deaths in Latin America are caused by some neoplasm, of which breast cancer accounts as the main responsible [15,16].

In 2003/2005, the National Health Foundation (FUNASA) reported the diagnosis of 45 cases of breast cancer in indigenous people in Brazil [17]. In addition, according to Freitas et al. (2015) [12], in 2000 and 2010, there were fifteen cases of death from breast cancer in Native Americans (13.31% in 2000 and 5.01% in 2010). There is a lack of evidence in the literature about current information concerning breast cancer in the population of the Americas, probably due to an underreporting of cases in local communities [8,14]. Furthermore, the traditional populations of Latin America constitute a complex study group due to their human history of miscegenation, which confers them with high levels of interpopulation genetic diversity [10].

The detection of germline mutations is essential for clinical control and early screening for breast cancer. Nevertheless, the accumulated knowledge about mutation profiles in the *BRCA1* and *BRCA2* genes in the Brazilian population, particularly in American Amazonian populations, is insufficient [18]. The study population of the Online Archive of Brazilian Mutations (ABraOM) database, which is composed of Brazilian individuals from the southeast of the country, who form a very heterogeneous population with higher degrees of European ancestry, followed by African ancestry and only 10% of Amerindian ancestry [9], does not represent a good resource for the care of individuals from the isolated and homogeneous Amerindian groups from the Brazilian Amazon.

Previous studies report a significant heterogeneity in the mutation burden of the *BRCA1* and *BRCA2* genes between different countries and even among regions of the same country (for example, in Brazil), which impairs the effectiveness and applicability of the use of already existing panels in Amerindian and Brazilian populations [18,19].

Therefore, our study aimed to characterize the molecular profile of the *BRCA1* and *BRCA2* genes through the analysis of the exome of Amerindian populations from the Brazilian Amazon region, to describe variants that can be associate with the risk of breast cancer, and to compare the results with publicly available data about global populations.

This is the first study to investigate the *BRCA1* and *BRCA2* genes in Amazonian Amerindians, a genetically distinct population that is unrepresented in genomic investigations. The Amerindian ancestry contribution in Brazil has an average of 17%, while in the Amazon region, this level of admixture is approximately 30%, standing out as the region with the greatest Amerindian genetic contribution of Brazil [20,21].

Of the 41 variants found in the exome analysis of the *BRCA1* and *BRCA2* genes, our study analyzed 15 that could be potentially related to the risk of developing breast cancer. Among the variants investigated, ten of them show a moderate impact, of which nine are classified as missense and one as synonymous; the five remaining variants had a modifier impact, being three intronic, one in the 3′-UTR region and one in the 5′-UTR region.

Among the eight single nucleotide polymorphisms (SNP) of the *BRCA1* gene, the rs799923, rs1799966 and rs799917 have been associated with the increased risk of BC in Asians [22,23]. In contrast, the rs16942 and rs1799949 variants have been associated with the predisposition to BC in the African population [24,25]. Other investigations linked the rs16941 polymorphism with susceptibility of BC in Caucasians [24]. The rs799917 and rs12516 polymorphisms in the *BRCA1* gene have been little explored in genetic association studies [26,27].

Regarding the seven SNPs of the *BRCA2* gene, there is an association of the development of BC with the rs115771651, rs766173, and rs1799943 variants in the Asian population [28,29] and rs1799944 and rs1799943 polymorphism in Caucasians [30,31]. The other three polymorphisms (rs11571707, rs11571769, rs144848) of the *BRCA2* gene evaluated here are related to hereditary breast cancer [32,33]. All these three variants had high allelic frequencies in the Amerindian populations in comparison to the global populations analyzed.

The rs11571707 showed the greatest difference between the allele frequency of the IND individuals (32.03%) and the majority of the global populations, which had a frequency of less than 1%, except the AMR that showed a frequency of 14%. However, the AMR population of the 1000 Genomes Project is composed mainly of Latinos, who present levels of Amerindian miscegenation [11]. According to Solano et al. (2012), the rs11571707 variant is often associated with hereditary breast and ovarian cancer syndrome (HBOC) in Latin Americans [34,35]. As observed in Table 3, the rs11571707 showed 12.5% (8 out of 64) IND individuals with homozygous mutant genotype.

Similarly, the rs144848 had a significantly higher frequency in the Amazonian Amerindian peoples (50.78%) when compared to Americans (30.5%), Europeans (28.2%), South Asians (35.5%), East Asians (27.3%), and, particularly, Africans (12%). This variant has been pointed to act as a factor that confers a moderate risk for the development of BC [36,37,38]. Studies have already observed a relationship between the mutant homozygous genotype and the increased risk of breast cancer in women under 60 years of age [36] and family members of breast cancer patients [39]. As shown in Table 3, 25% (16 out of 64) IND individuals were homozygous mutant for the rs144848 variant.

Furthermore, the rs11571769 showed a frequency of 11.1% in the IND group of this study in contrast to a frequency of 3% in AMR, 1.4% in SAS, less than 1% in both AFR and EUR (the variant allele was not reported in the EAS populations). According to a study conducted with patients with HBOC patients in Brazil, the rs11571769 was identified as a predictor of pathogenicity for BC development [40,41]. Our analyses did not report any homozygous mutant individual for the rs11571769 variant in the IND group.

The Amerindians from the Amazon region have a unique genetic profile because of stochastic processes resulting from a long process of geographic isolation and inbreeding [42]. Additionally, some small groups migrated from settled areas to uninhabited territories, giving rise to the first indigenous communities [36]. The genetic drift effect in these populations was even more driven by the small population size and inbreeding relationships, resulting in higher numbers of deleterious homozygous genotypes in the Native American populations [37,38].

The MDS analysis resulted in the formation of a single group formed by AMR and IND, which was caused by their genetic similarity (Fst = 0.03669). The population of Americans is known for their trichotomous genetic admixture with ancestral contributions of European colonists, African immigrants, and Native Amerindians [43,44]. Thus, since the Amerindians were one of the forming group of the Latin Americans, they have similarities in their genomic profile, including the *BRCA1* and *BRCA2* genes, a fact that can be extremely relevant in the likely high frequency of variants linked to breast cancer in these populations [45,46].

In contrast, the comparative analysis of the *BRCA1* and *BRCA2* variants in the AMR population showed 10 polymorphisms with significant statistical differences to the IND population. The sample of the American population of the 1000 Genomes Project includes several countries of Latin America, such as Mexico, Peru, Colombia, and Puerto Rico [11]. Although there is a degree of genetic similarity between AMR and IND populations, the genomic profiles of these countries indicate considerable heterogeneity in terms of Amerindian ancestry contribution, resulting from the different historical aspects of their formation and their degree of genetic admixture, which may explain the differences found in the *BRCA1* and *BRCA2* gene variants [10,47,48,49].

## 5. Conclusions

The exome analysis of the *BRCA1* and *BRCA2* genes in the Amazonian Amerindian populations reported 15 polymorphisms potentially capable of increasing susceptibility to breast cancer. Among them, we highlight the rs11571707, rs144848, and rs11571769 variants of the *BRCA2* gene due to their association with the hereditary breast cancer onset. This study may help the establishment of future public policies for early breast cancer screening in Amerindian populations and also Brazilian individuals with high levels of Amerindian ancestry contributions.

## Figures and Tables

**Figure 1 genes-12-00142-f001:**
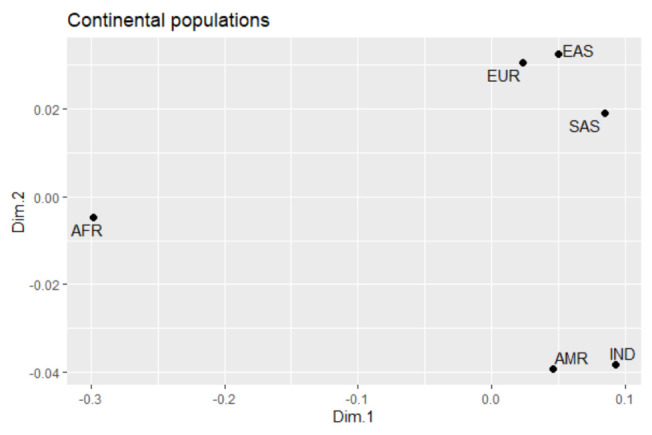
Multidimensional scaling plot illustrating the grouping of ethnic populations according to the genetic profile of the 15 variants in the *BRCA1* and *BRCA2* genes.

**Table 1 genes-12-00142-t001:** Description of the variants in the *BRCA1* and *BRCA2* genes in the Amerindian individuals (IND) and continental populations (African (AFR), American population (AMR), East Asian (EAS), European (EUR), and South Asian (SAS)) described in the 1000 genomes database.

Gene	SNP ID	Region	Change in Nucleotide	Impact Predicted by SNPeff	Minor Allele Frequencies					
					IND	AFR	AMR	EAS	EUR	SAS
*BRCA1*	rs799923	Intronic	G > A	Modifier	0.016	0.0389	0.0845	0.0007	0.2299	0.1797
*BRCA1*	rs16941	CDS	T > C	Moderate	0.453	0.1797	0.3083	0.3777	0.3254	0.4996
*BRCA1*	rs16942	CDS	T > C	Moderate	0.453	0.2358	0.3141	0.3827	0.3259	0.4999
*BRCA1*	rs1799966	Other ^a^	T > C	Moderate	0.453	0.2397	0.3138	0.3805	0.3268	0.5001
*BRCA1*	rs12516	3UTR	G > A	Modifier	0.24	0.3012	0.3433	0.4286	0.3841	0.4839
*BRCA1*	rs799905	Intronic	G > C	Modifier	0.4754	0.7789	0.4468	0.4403	0.3796	0.5116
*BRCA1*	rs1799949	Other ^a^	G > A	Moderate	0.4531	0.2364	0.3138	0.3783	0.3252	0.5002
*BRCA1*	rs799917	CDS	G > A	Moderate	0.4841	0.8193	0.3435	0.378	0.3341	0.5285
*BRCA2*	rs11571769 ^b^	CDS	G > A	Moderate	0.1111	0.0012	0.0349	0	0.0044	0.014
*BRCA2*	rs144848 ^b^	CDS	A > C	Moderate	0.5078	0.1249	0.3049	0.2728	0.2818	0.3558
*BRCA2*	rs1799943	5UTR	G > A	Modifier	0.0937	0.1005	0.1895	0.3805	0.2581	0.2829
*BRCA2*	rs11571651	Intronic	G > T	Modifier	0	0.0238	0.0633	0.0994	0.0335	0.1157
*BRCA2*	rs1799944	CDS	A > G	Moderate	0.1406	0.0427	0.0672	0.1002	0.035	0.1151
*BRCA2*	rs11571707 ^b^	CDS	T > C	Moderate	0.3203	0.0026	0.1409	0.0036	0.0003	0.0018
*BRCA2*	rs766173	CDS	A > C	Moderate	0.1379	0.024	0.0667	0.1002	0.035	0.1159

^a^ NEXT_PROT (modified-residue:phosphoserine); ^b^ variants related to hereditary breast and ovarian cancer syndrome (HBOC) accordingly to ClinVar.

**Table 2 genes-12-00142-t002:** Comparison between the allelic frequency of the study population Indian (IND) and continental populations (African (AFR), American (AMR), East Asian (EAS), European (EUR), and South Asian (SAS)) described in the 1000 Genomes database.

Gene	SNP Id	IND vs. AFR *	IND vs. AMR *	IND vs. EAS *	IND vs. EUR *	IND vs. SAS *
*BRCA1* ^a^	rs799923	0.5186	**0.0409**	0.05706	**1.705 × 10^−6^**	**9.866 × 10^−5^**
*BRCA1*	rs16941	**5.371 × 10^−7^**	**0.01994**	0.2428	**0.03282**	0.5307
*BRCA1*	rs16942	**1.614 × 10^−4^**	**0.02113**	0.2474	**0.03306**	0.5307
*BRCA1*	rs1799966	**1.96 × 10^−4^**	**0.02108**	0.2449	**0.04436**	0.5307
*BRCA1*	rs12516	0.663	0.3983	**0.06793**	0.1548	**0.01586**
*BRCA1* ^a^	rs799905	**4.308 × 10^−7^**	0.6986	0.6052	0.1459	0.6085
*BRCA1*	rs1799949	**1.657 × 10^−4^**	**0.02108**	0.2433	**0.03272**	0.5307
*BRCA1*	rs799917	**1.877 × 10^−9^**	**0.02391**	**0.0917**	**0.01618**	0.5305
*BRCA2*	rs11571769 ^b^	**4.044 × 10^−11^**	**0.00679**	**8.917 × 10^−14^**	**1.71 × 10^−8^**	**3.626 × 10^−5^**
*BRCA2* ^b^	rs144848	**1.567 × 10^−13^**	**9.83 × 10^−4^**	**6.84 × 10^−5^**	**1.429 × 10^−4^**	**0.01323**
*BRCA2*	rs1799943	1	0.05333	**3.735 × 10^−7^**	**0.00149**	**3.912 × 10^−4^**
*BRCA2*	rs11571651	1	0.3988	0.1629	1	0.1031
*BRCA2*	rs1799944	**0.001716**	**0.03866**	0.2925	**3.891 × 10^−4^**	0.5529
*BRCA2* ^a,b^	rs11571707	**<2.2 × 10^−16^**	**0.0002095**	**<2.2 × 10^−16^**	**<2.2 × 10^−16^**	**<2.2 × 10^−16^**
*BRCA2*	rs766173	**8.935 × 10^−5^**	0.0563	0.3749	**9.218 × 10^−4^**	0.5394

^a^ intronic variant; ^b^ variants related to hereditary breast and ovarian cancer syndrome (HBOC); * *p*-value obtained by Fisher’s exact test; bold: significant result (*p*-value ≤ 0.05).

**Table 3 genes-12-00142-t003:** The distribution of mutant alleles among the indigenous groups.

Amerindian Group	N. of Individuals	Number of Variant Alleles		
		rs11571769	rs11571707	rs144848
Asurini of the Xingu	5	2	4	4
Arara	7	0	8	8
Araweté	6	4	4	4
Asurini of the Tocantins	16	0	5	23
Awa-Guajá	8	1	4	6
Kayapó/Xikrin	2	1	1	1
Zo’é	5	4	6	6
Wajãpi	10	1	8	10
Karipuna	1	0	0	0
Phurere	1	0	1	1
Munduruku	1	1	1	1
Juruna	2	0	0	0

## Data Availability

The dataset used in this study is publicly available. The name of the repository and accession number(s) can be found at https://doi.org/10.6084/m9.figshare.13623989.v1.

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
