# Peer review of "Identification of Variants (rs11571707, rs144848, and rs11571769) in the BRCA2 Gene Associated with Hereditary Breast Cancer in Indigenous Populations of the Brazilian Amazon"

_genes, 2021, doi:10.3390/genes12020142_

Round 1

Reviewer 1 Report

The study titled “Identification of variants (rs11571707, rs144848, and 2 rs11571769) in the BRCA2 gene associated with hereditary breast cancer in indigenous populations of the Brazilian Amazon” by Dobbin et. al. is interesting but has very limited scientific conclusions to offer. I have following major concerns.

  • The sample size is too small (only 64 amerindians) to make a valid conclusion. Also it is not mentioned anywhere that these 64 amerindians were females, their age or any present condition of cancer.
  • The sample size of other five populations used for comparison from 1000 genome project is not included.
  • Does authors have any rationale that can explain why variants showing significant difference belongs to BRCA2 gene rather than any variant in BRAC1 gene?
  • What do we know about BRAC2 variants - rs11571707, rs144848, and 2 rs11571769 from literature or from genetic analysis of other populations?
  • How this study will help in establishment of future public polices for early cancer screening in Amerindian population?

Reviewer 2 Report

The study by Dobbin et al. examines frequencies of selected BRCA1 and 2 variants in Amazonian Amerindians, as a first step towards the elucidation of their possible contribution to breast cancer. The topic is of importance as Amazonian Amerindians are usually under-represented in studies, which can lead to lower quality of care. The authors conclude that three BRCA2 variants are represented at higher frequencies among Amazonian Amerindians than in five populations represented in 1000 Genomes. The paper is well written but lacks some important information. It seems desirable to address the following comments and inquiries.

-In the description of the population:

Lane 65: Change “The Amerindians” to “They” or remove period and change to “who represent… “

Does the report on ancestry comes from self-identification by the participants or from ancestry informative genetic markers? If these markers are not available/known, please indicate.

Are some participants closely related?

What is the status of the participants in regard to (breast) cancer or (breast) cancer in the family? How can this study inform on association of some variants with breast cancer variants (as stated in lanes 194-197) when the status is not described?

It seems that the exome data have been analyzed for variation in NUTDT15 in PMID: 32294118 (cited as ref 45 at the end of the paper). If so, add reference to that paper earlier.

-The variants should be described with more details.

For example, for rs11571707, the following information should be in the main text

NM_000059.4:c.7469T>C; NP_000050.3:p.Ile2490Thr

For rs144848 and rs11571769, it could be noted that they are polymorphic. Bold is used to describe the variant seen in this study.

rs144848 A>C, A>G

NM_000059.4:c.1114A>C, NP_000050.3:p.Asn372His

NM_000059.4:c.1114A>G, NP_000050.3:p.Asn372Asp

rs11571769 G>A, G>T

NM_000059.4:c.8851G>A NP_000050.3:p.Ala2951Thr

NM_000059.4:c.8851G>T NP_000050.3:p.Ala2951Ser

-The study is limited to the application of one software (SNPeff) to evaluate the variant potential deleterious effect and one database (1000 Genomes) to evaluate the distribution in populations.

Larger databases are available (ex: gnomAD gnomAD (broadinstitute.org)). Why was 1000 Genomes selected? It may be of interest to check additional databases for additional information. For example, gnomAD reports the highest representation of rs144848 in Ashkenazi Jewish (0.3566 over 0.2733 in total population).

Surprisingly, the study does not mention the web‐based public database ABraOM (Online Archive of Brazilian Mutations). The authors should incorporate data from this database (including for the MDS analysis) or justify why they do not consider it. (A quick analysis seems to indicate that the frequencies of the rs11571769, rs11571707 and rs144848 in ABraOM are much lower than what is reported for IND (0.011, 0.021 and 0.261))    

-The populations of reference (subsets of 1000 Genomes) should be better described. For each variant, what is the size of the various populations (total number of alleles, number of observed mutant alleles). Are the populations large enough for statistical significance, given the frequencies of the variants?

AMR needs to be defined in a consistent way rather than as Latin (lane 35) or American population (lanes 135, 159, 244). Does AMR really refer to a continental population (lane 122, Title of Figure 1, Table 2,…) or an admixed group as reported in gnomAD (Latino/Admixed American)? (In the United States, individuals with roots from Latin America are often classified as Hispanic and/or Latino). See for example: Towards a fine-scale population health monitoring system (biorxiv.org). Lanes 222-223 and 250-253 should be tighter together and supported by reference(s).

-Considering the frequencies reported in Table 2, it seems that the authors detected 14, 20 and 41 mutant alleles in rs11571769, rs11571707 and rs144848 respectively. (This information should be available to the reader without the need to calculate it). Homozygous individuals for the three variants are reported in gnomAD. Were some individuals homozygous in the IND set? How were the mutant alleles distributed among the 12 ethnic groups listed in M&M? Several rs in BRCA1 have the same frequencies in IND (ex: rs16941, 16942, 1799966, 1799949). Are these observed in the same individuals? Are they components of an haplotype?  

Note that the gnomAD database reports on unbalance in frequencies in male and female for rs11571707 in some populations (for example: 0.02204 in female, 0.01383 in male for total population but no bias in Latino/Admixed American). Was any sex-bias observed in the IND set?

-Lanes 167-169: the association of the variants with hereditary breast cancer should be better documented. For example, the database LitVar from NCBI LitVar - NCBI - NLM - NIH compiles 13, 20 and 149 publications for rs11571769, rs11571707 and  rs144848 respectively. These publications should be analyzed, especially as some, that are not currently referenced, report the variants in Brazilian populations

Note, in particular, in PMID: 31658756:

Page 5: …Some other BRCA1/2 variants are observed in more than one LA country (Table 3), like the pathogenic variant BRCA2 c.2808_2811del variant reported in seven countries (Argentina, Brazil, Colombia, Mexico, Peru, Uruguay, and Venezuela) and the pathogenic BRCA1 variants c.68_69delAG and c.211A>G observed in six countries. Remarkably, the BRCA2 variant c.7469T>C classified as benign in ClinVar was observed in HBC cases in Argentina, Brazil, Colombia, Cuba, Mexico, Uruguay, and Venezuela, prompting a reconsideration of its reclassification.

What is the current ClinVar classification of rs11571769, rs11571707 and rs144848? Considering that rs11571707 is reported at a frequency of 32% in the IND population, should its classification be reconsidered? What is the meaning of the data 1)-for the Amazonian Amerindian population, 2)-for the global population? A suggestion is for the authors to consider PMID: 28111427 for the description of BRCA1 p.Leu1780Pro as a founder mutation in Koreans and the information on interpretation of variants with high frequencies in a particular ethnic group. If possible, discuss whether any of the described variants could be a (pathogenic) founder mutation (and what it would take to test/validate, for example in terms of samples to analyze).

-Around lane 245, PMID: 27741520 could be added as a reference as Figure 3 shows ancestry contribution (including Amerindian) of BRCA1 c.5266dupC mutated patients.  

Page 80475…The c.5266dupC mutation represents about 98–99% of the mutations found in Ashkenazi Jews, however it is not restricted to these populations. It has been reported as having high prevalence in several countries, mainly from Central and Eastern Europe [33]. Among the 18 patients in our study who are carriers of this mutation, although we could not assess the local ancestry of the locus containing the BRCA1 c.5266dupC mutation, the European ancestry profile was prevalent in 94.4% of cases (17/18 families). For 83.3%, the European contribution was higher than 69%. In one case, the main contribution was the Amerindian followed by the European, demonstrating, once again, the wide miscegenation present in the Brazilian population…

Round 2

Reviewer 1 Report

The authors responded all my critiques. 

Author Response

We kindly thank the Reviewer for his/her comments and we reinforce that the main text has undergone a review and edition of the English language. 

Reviewer 2 Report

The manuscript has been improved. The following comments still need to be addressed.

-Lane 69: Replace at PMID: 32294118 [10] by [10].

-Lane 170: Remove associated with hereditary breast cancer as no reference is given. (Keep this element for the discussion -starting from lane 213)

-Lane 175-180: Replace allels by alleles

-The allele counts reported in lanes 173-181 do not match the mutant frequencies (ex: 32.03% - 20 mutant allels is incorrect. If 20 alleles are reported in a population of 64, the frequency is 20/128 or 15.6%).

Upon reviewing the Table provided in response to point 11, it appears that the values should be corrected as:

The rs11571707 (NM_000059.4:c.7469T>C; NP_000050.3:p.Ile2490Thr) showed the greatest allele frequency in the study subjects (32.03% - 42 mutant alleles) in contrast to the world populations (except the AMR) where the frequencies are less than 1%. Similarly, the rs144848 (NM_000059.4:c.1114A>C, 176 NP_000050.3:p.Asn372His) also had a significantly high frequency in indigenous people (50.78% - 64 mutant alleles), whereas the other populations showed a mean frequency lower than 27%.

-Lane 190: change American so that it is not misunderstood as from the USA (by US readers).

-Some information provided to the reviewer should be incorporated in the text.

  • Responses to point 3 (no family relationships) and 4 (no breast cancer or breast cancer in the family): to be included in study population
  • The table presented as response to point 11 should be incorporated in the main text (around lane 172) as Table 3. Indicate under parentheses how many individuals of each population are homozygous. In lane 236, indicate: as shown in Table 3, 25% (16 out of 64) IND individuals were homozygous mutant for rs144848.
  • The response given to point 8 on the ABraOM database may be incorporated in the discussion around lane 194. A suggestion for addition is as follow:

The study population of the ABraOM (Online Archive of Brazilian Mutations) database, which is composed of Brazilian individuals from the northeast of the country, who form a very heterogeneous population with higher degrees of European ancestry, followed by African ancestry and only 10% of Amerindian ancestry [Santos, current ref 19], does not represent a good resource for the care of individuals from the isolated and homogeneous Amerindian groups from the Brazilian Amazon.

-As a comment to the responses (7, 9, 10) addressing the need to better describe the 1000 Genomes database, it is expected that the manuscript presents a stand-alone study and that the reader does not need to access any website (which content may change with updates) to get more information. While the text has been improved with the additions of lanes 77-78, it seems important to consider the description of the 1000Genomes in the Supplementary data of the current reference 38 - Section 2.1.1 Choice of populations included in the project. It is recommended to add a paragraph that describes the reference populations and reinforces the complexity of the studies involving peoples from the Americas.

A suggestion for addition after lane 78 is as follow:

As indicated in [current 38], 1- for the samples with ancestry from Europe, East Asia, and South Asia, populations across the geographic range had about 1% FST; 2- the populations from Africa are related to the Yoruba and are therefore not comprehensive within Africa; 3- for populations in the Americas, the samples are from two populations with primarily African and European ancestry (People with African Ancestry in Southwest USA (ASW) and African Caribbean in Barbados (ACB)) and four populations (People with Mexican Ancestry in Los Angeles, CA, USA (MXL), Colombians in Medellin, Colombia (CLM), Puerto Ricans in Puerto Rico (PUR), Peruvians in Lima, Peru (PEL)) with a wide range of European, African, and indigenous American ancestry chosen to represent the wide variation in ancestry proportions observed in North, Central, and South America.

-Considering that the description of 1000 Genomes is now substantial, the title “2.1. Study population” may be changed to “2.1. Study and reference populations.”
